

# Contribution of prosthetic treatment considerations for dental extractions of permanent teeth

Miguel Ángel Fernández-Barrera[1], Carlo Eduardo Medina-Solís[1,2], Juan Fernando Casanova-Rosado[3], Martha Mendoza-Rodríguez[1], Mauricio Escoffié-Ramírez[4], Alejandro José Casanova-Rosado[3], José de Jesús Navarrete-Hernández[1] and Gerardo Maupomé[5]

[1] Academic Area of Dentistry of Health Sciences Institute, Autonomous University of Hidalgo State, Pachuca, Hidalgo, Mexico
[2] Advanced Studies and Research Center in Dentistry "Dr. Keisaburo Miyata," School of Dentistry, Autonomous University State of Mexico, Toluca, Estado de Mexico, Mexico
[3] Faculty of Dentistry, Autonomous University of Campeche, Campeche, Mexico
[4] Faculty of Dentistry, Autonomous University of Yucatan, Merida, Yucatan, Mexico
[5] School of Dentistry, Indiana University-Purdue University at Indianapolis, Indianapolis, IN, United States

Corresponding author
Carlo Eduardo Medina-Solís,
cemedinas@yahoo.com

## ABSTRACT

**Background.** Tooth loss is an easily identifiable outcome that summarizes a complex suite of factors in an individual's history of dental disease and its treatment by dental services over a lifetime. Assessment of overall tooth loss data is essential for epidemiologically evaluating the adequacy of dental care provided at a systems level, as well as for placing in context tooth loss for non-disease causes. For example, when derived from prosthetic treatment planning, the latter may unfortunately lead to some teeth being extracted (pulled) for the sake of better comprehensive clinical results. The objective of the present manuscript was to identify the contribution to overall tooth loss, by extraction of permanent teeth because of prosthetic treatment reasons.

**Material and Methods.** A cross-sectional study included sex, age, total number of extractions performed by subject, sextant (anterior vs. posterior), group of teeth (incisors, canines, premolars and molars), upper or lower arch, and the main reason underlying extraction (extraction for any reason vs. prosthetic treatment), in patients 18 years of age and older seeking care at a dental school clinic in Mexico. A multivariate logistic regression model was generated.

**Results.** A total of 749 teeth were extracted in 331 patients; 161 teeth (21.5% of total) were extracted for explicit prosthetic treatment indications. As age increased, the likelihood of having an extraction for prosthetic reasons increased 3% (OR = 1.03, $p < 0.001$). Women (OR = 1.57, $p < 0.05$) were more likely to be in this situation, and molars (OR = 2.70, $p < 0.001$) were most at risk. As the total number of extractions increased, the risk of having an extraction for prosthetic reasons decreased (OR = 0.94, $p < 0.05$).

**Conclusions.** A significant amount (21.5%) of the extractions of permanent teeth were performed for prosthetic reasons in this dental school clinical environment; age, sex, type of tooth, and the total number of extractions moderated such pattern.

## INTRODUCTION

According to *Marcenes et al.* (*2014*), *Kassebaum et al.* (*2014a*) and *Kassebaum et al.* (*2015*) oral conditions have remained prevalent worldwide. Untreated caries (tooth decay) in permanent teeth is the most prevalent condition as evaluated in the landmark "Global Burden of Disease 2010 Study," while severe periodontitis (gum disease), and untreated caries in deciduous teeth were the 6th and 10th most prevalent conditions affecting 11% and 9% of the global population, respectively. Severe tooth loss was the 36th most frequent situation with a global estimate of 2%. Dental caries manifests as a continuum of disease states of increasing severity and tooth destruction. These can range from sub-clinical, asymptomatic changes in tooth structure to carious lesions with extensive pulpal involvement (*Kassebaum et al.*, *2014a*). Moreover, periodontal diseases are chronic disorders affecting the tissues supporting the teeth. Inflammatory events associated with loss of connective tissue also lead to resorption of alveolar bone support (*Armitage & Research, Science and Therapy Committee of the American Academy of Periodontology*, *2003*; *Greenwell, Committee on Research, Science and Therapy. American Academy of Periodontology*, *2001*). Caries and destructive periodontal diseases are major oral public health problems and often quoted to be the leading causes of permanent teeth extraction (pulling teeth) (*Haseeb, Ali & Munir*, *2012*; *Jafarian & Etebarian*, *2013*; *Saheeb & Sede*, *2013*; *Lee et al.*, *2015*).

Tooth loss is a multifactorial and complex outcome that reflects diverse circumstances of the individual's history of dental disease and its treatment with dental services over a lifetime (*Kassebaum et al.*, *2014b*). Tooth loss has been proposed as a negative indicator of oral health: various international oral health-related agencies have established global oral health goals for the year 2020 (*Hobdell et al.*, *2003*) that include preservation of dentition as one of the most important goals of preventive dentistry. Despite progress in technical procedures, tooth extraction is one of the most widely performed procedures in dentistry today in many parts of the world (*Lesolang, Motloba & Lalloo*, *2009*; *Alomari, Khalaf & Al-Shawaf*, *2013*). While assessment of tooth loss data is essential for evaluating the adequacy of dental care services provided across diverse locations in the world, it remains just as important to place in context why tooth loss happens—in particular when prosthetic treatment planning inevitably implies selective tooth extraction. Simply put, some teeth are extracted for the greater good of the mouth through treatment planning of dental services despite the fact that their survival as an individual tooth could have taken place (*Alomari, Khalaf & Al-Shawaf*, *2013*). Placing dental prostheses is a common dental course of treatment aiming to replace missing teeth; such rehabilitation of function and appearance may be attained using fixed or removable prostheses, partially replacing teeth or completely (full dentures). Treatment planning and management include diagnostic evaluation of whether teeth still in the mouth can be safely thought to be stable in their survival over time (therefore optimizing the likelihood of prostheses performing adequately), or if some of those teeth should be selectively eliminated and replaced through a prosthetic substitution. The guiding principle is that some teeth may need to be sacrificed for the sake of better prosthesis design or function. Factors often incorporated to this stage of treatment planning include location of tooth in dental arch, periodontal status, root-to-crown ratio of alveolar bone support,
need for and likelihood of success of endodontic treatment, interocclusal relationship, and aesthetic condition and relative contribution to overall appearance (*Davarpanah et al.*, *1998*; *Freitas-Júnior & Silva*, *2012*).

An accurate understanding of the relative contribution of prosthetic reasons to the extraction of teeth ought to differentiate actual reasons leading to overall tooth loss prevalence. Only a few reports in the world literature have made this distinction, and suggested a wide range of prevalence for tooth extractions undertaken for prosthetic reasons: from 23.2% (*Abreu et al.*, *1998*) to 3.4% (*Rubiños-López et al.*, *2008*). The objective of the present study was to add to such body of literature, through quantifying the contribution of prosthetic treatment planning as the reason for extracting permanent teeth, taking into account demographic factors.

## MATERIAL AND METHODS

### Study location

A cross-sectional study was undertaken on the clinical records of patients seeking care at dental school clinics of the Autonomous University of Hidalgo State (UAEH), in Central Mexico. The dental health care system in Mexico is a mixed and fragmented health system composed of public services and social security efforts supplied by public institutions, third party payment systems, and private carriers. The overwhelming majority of services are delivered under a fee-for-item, out-of-pocket scheme run by largely unregulated dental professionals and dental market. The public health sector is responsible for a small, essentially undetermined and largely fluid set of services that are almost restricted to the urban settings. In contrast, dozens of dental school clinics in numerous public and private universities offer dental services to the population, provided by students under close faculty supervision, at much-reduced prices (*Pérez-Núñez et al.*, *2006*). While not an organized, distinct clinical care system, services delivered in dental school clinics constitute a significant portion of dental care services available to the open population.

### Design and study population

Part of the methodology has been previously published (*Medina-Solís et al.*, *2013*; *Medina-Solís et al.*, *2014*). No sampling was performed because we enrolled all consecutive patients seen for uncomplicated tooth extractions in one calendar year (2009). Clinical examinations, taking medical/dental histories, and periapical radiographs are routinely performed on all patients scheduled for tooth extractions. The extractions incorporated to the present study ($n = 749$ in 331 patients) were performed under local anaesthesia by senior dental students under the supervision of clinical faculty. Inclusion criteria were: (a) patients of either sex, (b) 18 years of age or older, and (c) with complete clinical data available, as described above.

### Variables

Independent variables included patient age, which was divided into three groups: $0 = 18$–44 years of age, $1 = 45$–59 years, and $2 = \geq 60$ years; sex: $0 =$ men and $1 =$ women; arches: $0 =$ upper and $1 =$ lower; sextant: $0 =$ anterior and $1 =$ posterior; type of tooth: $0 =$ incisors,

1 = canines, 2 = premolars, and 3 = molars. The dependent variable was the reason why the extraction was performed; it was coded as 0 = any other clinical reason, or 1 = extraction for prosthetic reasons. The latter was supported by analyses of dental casts and radiographs evaluated by clinical faculty. Teeth scheduled for extraction for prosthetic reasons may had also been carious or have periodontal problems; however, categories for tooth extraction were based on the clinical notes and the treatment planning considerations entered. Even teeth that were seemingly healthy may have been scheduled for extraction as a part of a treatment plan for a dental prosthesis (fixed, removable, or full).

## Statistical analysis

Statistical analysis consisted of a description of the variables according to the scale of measurement. For bivariate and multivariate analysis, we used binary logistic regression. We fitted a multivariate model to estimate the strength of association between our dependent variable and the independent variables, which is expressed as odds ratios (OR) with 95% confidence intervals (95% CI). While we also reported the $p$ values that were considered statistically significant ($p$ value $< 0.05$), to construct the final model we followed standard guidelines by adding variables that in the bivariate analysis had a statistical significance of $p < 0.25$ (*Hosmer & Lemeshow*, *2000*; *Bagley, White & Golomb*, *2001*). The test variance inflation factor (VIF) was used to analyse, and where appropriate, to avoid multi-collinearity between independent variables. In the final model, we used a specification error test (link test). The test first considered the link function of the outcome variable on the left-hand side of the equation. We chose the *logit* function (in logistic regression) as the correct function to use. Second, on the right-hand side of the equation, we assumed that we had parsimoniously included all relevant variables. The logit function is a linear combination of the predictors. After fixing the main effects, interactions were tested; none proved to be significant at $p < 0.15$. We finally characterized the overall fit of the model per standard recommendations (*Hosmer & Lemeshow*, *2000*; *Bagley, White & Golomb*, *2001*). All analysis used Stata 11.0 (StataCorp, College Station, TX, USA).

## Ethical approval

All procedures performed on human participants were in accordance with the ethical standards of the institutional and/or national research committees, and with the 1964 Helsinki declaration, its later amendments, or comparable ethical standards. The approval reference/number from the Institutional Ethical Review Committee of the UAEH: 34-2009-RAZONES EXTRACCIONES-CEMS-11. Formal approval was granted by the institutional ethical review committee of the UAEH. Data collected were anonymized from the patient's charts.

## RESULTS

A total of 749 teeth were extracted in 331 patients aged 18 and older during the study period. Details of descriptive analysis are shown in Table 1. Mean age was 48.7 ± 13.5. Most extractions were performed on women ($n = 487$; 65.0%). There were 418 (55.8%) extractions performed in the maxilla, and most were posterior teeth ($n = 484$; 64.6%);

**Table 1** Descriptive analysis of independent variables included.

| Variable | Mean ± sd |
|---|---|
| Age | 48.70 ± 13.54 |
| Extracted teeth[a] | 4.25 ± 3.63 |
|  | Frequency (%) |
| Sex |  |
| Men | 262 (35.0) |
| Women | 487 (65.0) |
| Arch |  |
| Upper | 418 (55.8) |
| Lower | 331 44.2) |
| Sextant |  |
| Anterior | 265 (35.4) |
| Posterior | 484 (64.6) |
| Tooth type |  |
| Incisors | 181 (24.2) |
| Canines | 84 (11.2) |
| Premolars | 154 (20.6) |
| Molars | 330 (44.0) |

**Notes.**
[a] Number of extractions undertaken per patient.

specifically, molars accounted for 44.0% ($n = 330$). The results show that 161 teeth (21.5%) were extracted for prosthetic reasons.

The results of the bivariate analysis (Table 2) were taken into account for the construction of the multivariate logistic regression model in Table 3. It showed that as age increased, the likelihood of having an extraction driven by prosthetic treatment planning increased 3% (OR = 1.03, $p < 0.001$). Women (OR = 1.57, $p < 0.05$) were more likely to have a dental extraction for such reasons. Molars (OR = 2.70, $p < 0.001$) were the teeth most likely to be extracted as part of a pre-prosthetic treatment plan. As the total number of extractions per patient increased, the chance of having them done for prosthetic reasons decreased (OR = 0.94, $p < 0.05$).

## DISCUSSION

The study in a dental school clinic situated in an emerging economy country such as Mexico showed that one fifth (21.5%) of teeth were extracted because of prosthetic indications. This is a considerable proportion of the teeth extracted overall. Our figures are similar to those found in other studies; we annotate here the findings relevant to countries broadly in the same level of economic development and/or similar availability of dental care services. For example, *Abreu et al.* (*1998*) found that prosthetic treatment planning extractions accounted for 23.2% of all extractions performed in a dental school clinic in Minas Gerais, Brazil. Similarly, *Kalauz, Prpić-Mehičić & Katanec* (*2009*) conducted a study in a dental school clinic in Zagreb (Croatia) and found that the prevalence of prosthetic extractions

**Table 2  Bivariate analysis between prosthetic extractions and independent variables.**

|  | Others | Prosthetic | OR (95% CI) | *p* value |
|---|---|---|---|---|
| Age | 48.31 ± 13.91 | 50.14 ± 11.98 | 1.01 (1.00–1.02) | 0.128 |
| Extracted teeth | 4.41 ± 3.79 | 3.66 ± 2.90 | 0.94 (0.89–0.99) | 0.021 |
| Sex |  |  |  |  |
| Men | 218 (83.2) | 44 (16.8) | 1[a] |  |
| Women | 370 (76.0) | 117 (24.0) | 1.57 (1.07–2.30) | 0.022 |
| Arch |  |  |  |  |
| Upper | 335 (80.1) | 83 (19.9) | 1[a] |  |
| Lower | 253 (76.4) | 78 (23.6) | 1.24 (0.88–1.76) | 0.220 |
| Sextant |  |  |  |  |
| Anterior | 223 (84.2) | 42 (15.8) | 1[a] |  |
| Posterior | 365 (75.4) | 119 (24.6) | 1.73 (1.17–2.55) | 0.006 |
| Tooth type |  |  |  |  |
| Incisors | 156 (86.2) | 25 (13.8) | 1[a] |  |
| Canines | 67 (79.8) | 17 (20.2) | 1.58 (0.80–3.12) | 0.185 |
| Premolars | 131 (85.1) | 23 (14.9) | 1.09 (0.59–2.02) | 0.770 |
| Molars | 234 (70.9) | 96 (29.1) | 2.56 (1.58–4.16) | 0.000 |

Notes.
[a] Reference category.

**Table 3  Multivariate model of logistic regression between prosthetic extractions and independent variables.**

|  | OR (95% CI) | *p* value |
|---|---|---|
| Age | 1.03 (1.02–1.05) | 0.000 |
| Extracted teeth | 0.94 (0.88–0.99) | 0.027 |
| Sex |  |  |
| Men | 1[a] |  |
| Women | 1.57 (1.05–2.33) | 0.027 |
| Tooth type |  |  |
| Incisors/canines | 1[a] |  |
| Premolars | 1.00 (0.57–1.76) | 0.993 |
| Molars | 2.70 (1.72–4.22) | 0.000 |

Notes.
[a] Reference category.
Goodness of fit test: Hosmer–Lemeshow chi$^2$(8) = 10.00, *p* = 0.2651.
*Linktest* (specification error test): predictor = 0.030; predictor$^2$ = 0.648.

was 18.7%, while *Alomari, Khalaf & Al-Shawaf* (*2013*) reported the proportion of teeth extracted for prosthetic indications was 15.9% in Kuwait University. However, other authors have reported different results; *Jafarian & Etebarian* (*2013*) described a frequency of 4.1% in an Iranian study, while *Rubiños-López et al.* (*2008*) in Spain showed the rate was 3.4% in publicly funded clinical services. These variations may be partly attributed to diverse methodological approaches, to different settings included (dental school clinics, hospitals, or dental offices), and/or to various response rates, as well as wide variation in ages

of the populations sampled (*Jafarian & Etebarian*, *2013*). Tooth extraction should ideally be the last resort alternative among dental treatment options, and clinicians should be careful in deciding whether a tooth—especially a healthy tooth—should ever be removed. In making this decision, the clinician needs information such as tooth type or the prosthetic plan intended. Ethical implications apply, as the best interest of the patient ought to be driving the decision to undertake an extraction for prosthetic reasons; in this context, prevalence of extractions, their reasons, and associated factors ought to be available to clinicians. Moreover, candid discussion of such scenario and factors during dental training may strengthen the dental education curriculum. Specifically, age appears important as *Jafarian & Etebarian* (*2013*) in Iran, *Abreu et al.* (*1998*) in Brazil, and *Rubiños-López et al.* (*2008*) in Spain reported that in older people, extractions for prosthetic reasons increased in relation to age. Unlike the findings in our study, *Jafarian & Etebarian* (*2013*) observed that prosthetic reasons for extractions were more common among men. That Iranian study also reported that posterior teeth were extracted more frequently for prosthetic reasons than anterior teeth (*Jafarian & Etebarian*, *2013*). But *Abreu et al.* (*1998*) in Brazil, *Kalauz, Prpić-Mehičić & Katanec* (*2009*) in Croatia, and *Alomari, Khalaf & Al-Shawaf* (*2013*) in Kuwait noted that anterior teeth were preferentially extracted for prosthetic purposes.

There are no unequivocal guidelines for prosthetic extractions, and even less clinical guidelines that may be considered appropriate to cut across the enormous assortment of clinical practices, evidence based indications, or costing/reimbursement models for clinical care in the highly diverse locations across the most developed to the least developed countries. There is considerable scope to inform clinical practices through future research.

There are some limitations to the present study that should be considered when interpreting the data, in particular when attempting to apply its message to other contexts. For example, the information was obtained from patients who sought care in dental school clinics; it may therefore be not representative of dental offices or other service outlets in Mexico. Adding strength to our design, however, is the fact that faculty helping students in planning and undertaking prosthetic treatment led to a measure of criteria standardization. This feature may not be available outside dental school clinics. This is an important consideration, as *Kassebaum et al.* (*2014b*) noted that tooth loss reflects underlying dental disease as well as patients' and dentists' attitudes, the dentist-patient relationship, the availability and accessibility of dental services, and the prevailing philosophies of dental care. It is crucial to keep in mind that our data pertains directly only to one clinical care outlet; however important may be the care afforded to the open public through dental school clinics in Mexico, it is a domain without systematic data collection or evaluation. Market share, clinical impact, and public health significance are poorly understood—just as they are in many similar environments in either industrialized, emerging, and less-developed countries.

## CONCLUSION

A significant amount (21.5%) of the extractions of permanent teeth were performed as part of a prosthetic treatment plan in an open population of Mexican adults seeking care at

a dental school clinic. Age, sex, type of tooth, and total number of extractions moderated such pattern. Overall prevalence of tooth extractions should be considered in light of these results, to place in context the various reasons why some teeth are ultimately lost and not simply attribute all tooth loss to clinical neglect or barriers in access to care. While extracting a tooth is generally considered to be the last alternative in dental treatment options, the influence of factors such as age, the type of tooth, or specific prosthetic indications must also be considered.

### Funding
The authors received no funding for this work.

### Competing Interests
The authors declare there are no competing interests.

### Author Contributions
- Miguel Ángel Fernández-Barrera and Carlo Eduardo Medina-Solís conceived and designed the experiments, performed the experiments, analyzed the data, contributed reagents/materials/analysis tools, wrote the paper, prepared figures and/or tables, reviewed drafts of the paper, accepted the final version.
- Juan Fernando Casanova-Rosado conceived and designed the experiments, contributed reagents/materials/analysis tools, wrote the paper, prepared figures and/or tables, reviewed drafts of the paper, accepted the final version.
- Martha Mendoza-Rodríguez and José de Jesús Navarrete-Hernández performed the experiments, contributed reagents/materials/analysis tools, wrote the paper, reviewed drafts of the paper, accepted the final version.
- Mauricio Escoffié-Ramírez and Alejandro José Casanova-Rosado contributed reagents/materials/analysis tools, wrote the paper, reviewed drafts of the paper, accepted the final version.
- Gerardo Maupomé conceived and designed the experiments, contributed reagents/materials/analysis tools, wrote the paper, reviewed drafts of the paper, accepted the final version.

### Human Ethics
The following information was supplied relating to ethical approvals (i.e., approving body and any reference numbers):

Institutional ethical review committee of the UAEH.

34-2009-RAZONES EXTRACCIONES-CEMS-11.

### Data Availability
The raw data has been supplied as Data S1.

## Supplemental Information

Supplemental information for this article can be found online at http://dx.doi.org/10.7717/peerj.2015#supplemental-information.

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
