# Peer review of "Contribution of prosthetic treatment considerations for dental extractions of permanent teeth"

_PeerJ, doi:10.7717/peerj.2015_

## Round 0.1 · original submission · Major Revisions

The design using patients treated at dental schools may make it difficult to draw any general conclusions. This needs to be considered in the conclusion too.

In addition, note Reviewer 1's concerns regarding greater clarity, which precluded a full review.

Reviewer 1 ·

Basic reporting

Introduction is in part unclear. Needs to be adjusted. Language is also vague in some parts. The text about mortality of tooth seems inappropriate.
Aim is also not completely distinct.

The paper lack a definition of the prosthetic extraction. How is it defined and applied to the material?

Experimental design

The chosen p-value för significance level has not been defined.

Validity of the findings

As the definition of the term "prosthetic extraction" is not defined, the results are not possible to validate.

Additional comments

I recommend revision of the paper in terms of language and clearity of the aims, introduction and definition of the term "prosthetic extraction".

·

Basic reporting

Thank you for an interesting manuscript. In my view, it needs revision before it could be considered to be published.

In general your English is quite correct, although you must read the text again and make some smaller changes. For example, look at page 2, line 33 in the abstract. In the material and methods section, page 5, line 95 you refer to a study published in 20014. On the same page, line 99, there is a sentence that repeats what you just have written. Please remove it. On page 5, line 100, you start the sentence with a number (749). Please write the number with letters when it is in the beginning of a sentence. On page 6, line 136, you write “extractions were performed in the maxillary teeth”. Do you mean “in the maxilla”?

The introduction is somewhat unbalanced as you discuss caries and periodontitis much more than the real subject for your study: prosthodontic considerations for tooth extractions. When you finally write about prosthodontic considerations you do it very briefly. I suggest that you go deeper into this later part, both in the introduction and in the discussion sections.

In the material and methods section, on page 5, you write that the patients were enrolled during a year (lines 95-96). Please specify that time period. Further, on the same page, line 100, you mention the number of extractions. Why not describe the number of patient as well, in these sections?

Why have you chosen statistical significance levels of p‹0.25 and p‹0.15 (pages 5 and 6), and why so extremely high levels? Why not use the more common p‹0.05 or less, especially as you on page 6, line 120, write that “many interactions were tested”? Please explain. Thoroughly

As written above, I suggest that you explore and discuss the prosthodontic considerations for extractions further. How has prosthodontic reasons for tooth extraction been discussed in earlier studies, and what is your opinion and definitions? Why not discuss the ethical considerations to be kept in mind when analyzing reasons for teeth removal?

Experimental design

In general, the design is easy to follow but as mentioned above it needs to be clarified further. Not least the choice of statistical significance levels must be explained. Also, the definition of prosthodontic reasons for tooth extractions must be explained further.

Validity of the findings

The validity of the findings depends of how thoroughly you describe the material and methods, and also how you discuss and explain why it is interesting to study prosthodontic reasons for tooth removal.

Additional comments

I my view, you must revise your manuscript at least when it comes to the introduction, material and methods and discussion sections. Please do.

---

## Round 0.2 · accepted · Accept

The paper is improved considerably.

·

Basic reporting

Your manuscript is now much more balanced. The language and aims are clearer and the definitions are more distinct.

Experimental design

No comments.

Validity of the findings

I my view, you discuss the validity of the findings and the limitations of the study in a much better way than in the previous version of your manuscript.

Additional comments

You have made a great improvement.